# SCAMPS: Synthetics for Camera Measurement of Physiological Signals

**Daniel McDuff**
Microsoft
Redmond, WA, USA

**Miah Wander**
Microsoft
Redmond, WA, USA

**Xin Liu**
UW
Seattle, WA, USA

**Brian L. Hill**
UCLA
Los Angeles, CA, USA

**Javier Hernandez**
Microsoft
Redmond, WA, USA

**Jonathan Lester**
Microsoft
Redmond, WA, USA

**Tadas Baltrusaitis**
Microsoft
Cambridge, UK

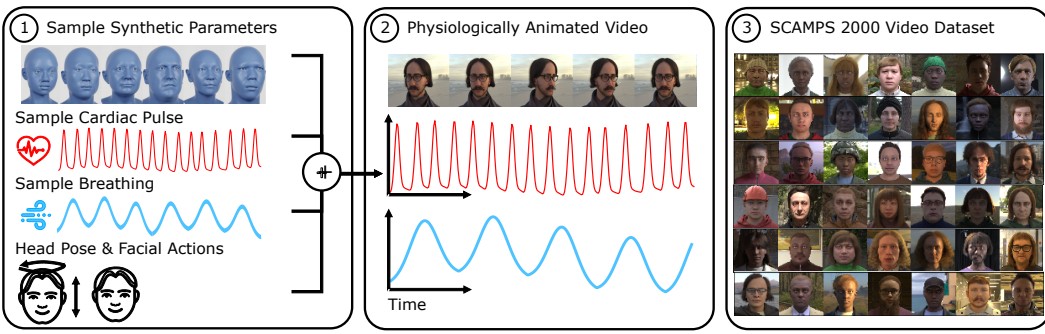

Figure 1: SCAMPS: A dataset of synthetic videos with aligned physiological and behavioral signals.

## Abstract

The use of cameras and computational algorithms for noninvasive, low-cost and scalable measurement of physiological (e.g., cardiac and pulmonary) vital signs is very attractive. However, diverse data representing a range of environments, body motions, illumination conditions and physiological states is laborious, time consuming and expensive to obtain. Synthetic data have proven a valuable tool in several areas of machine learning, yet are not widely available for camera measurement of physiological states. Synthetic data offer "perfect" labels (e.g., without noise and with precise synchronization), labels that may not be possible to obtain otherwise (e.g., precise pixel level segmentation maps) and provide a high degree of control over variation and diversity in the dataset. We present SCAMPS, a dataset of synthetics containing 2,800 videos (1.68M frames) with aligned cardiac and respiratory signals and facial action intensities. The RGB frames are provided alongside segmentation maps and precise descriptive statistics about the underlying waveforms, including inter-beat interval, heart rate variability, and pulse arrival time. Finally, we present baseline results training on these synthetic data and testing on real-world datasets to illustrate generalizability.

Project webpage: https://github.com/danmcduff/scampsdataset

36th Conference on Neural Information Processing Systems (NeurIPS 2022) Track on Datasets & Benchmarks.

Table 1: Summary of Public Camera Physiological Measurement Datasets.

| Dataset | Subjects | Videos | Gold Standard | Sub. Div. | Env. Div. | Free Access |
|---|---|---|---|---|---|---|
| MAHNOB [36] | 27 | 527 | ECG, EEG, BR | ✗ | ✗ | ✓ |
| BP4D [55] | 140 | 1400 | BP, AU | ✓ | ✗ | ✗ |
| VIPL-HR [28] | 107 | 3130 | PPG, HR, $SpO_2$ | ✗ | ✗ | ✓ |
| COHFACE [14] | 40 | 160 | PPG | ✗ | ✗ | ✓ |
| UBFC-RPPG [4] | 42 | 42 | PPG, PR | ✗ | ✗ | ✓ |
| UBFC-PHYS [27] | 56 | 168 | PPG, EDA | ✗ | ✗ | ✓ |
| RICE CamHRV [31] | 12 | 60 | PPG | ✗ | ✗ | ✓ |
| MR-NIRP [29] | 18 | 37 | PPG | ✗ | ✗ | ✓ |
| PURE [39] | 10 | 59 | PPG, $SpO_2$ | ✗ | ✗ | ✓ |
| rPPG [16] | 8 | 52 | PR, $SpO_2$ | ✗ | ✗ | ✓ |
| OBF [19] | 106 | 212 | PPG, ECG, BR | ✗ | ✗ | ✗ |
| PFF [15] | 13 | 85 | PR | ✗ | ✗ | ✓ |
| VicarPPG [43] | 20 | 10 | PPG | ✗ | ✗ | ✓ |
| CMU [8] | 140 | 140 | PR | ✓ | ✓ | ✓ |
| SCAMPS* | 2800 | 2800 | PPG, PR, BR, AU | ✓ | ✓ | ✓ |

ECG = Electrocardiogram waveform, EDA = Electrodermal activity, EEG. = Electroencephalogram waveforms, PPG = Photoplethysmogram waveform, BP = Blood pressure waveform, PR = Pulse rate, BR = Breathing rate, $SpO_2$ = Blood oxygenation, AU = Action Units.
* SCAMPS is the only synthetic dataset.

# 1 Introduction

Camera physiological measurement is a rapidly growing field of computer vision and computational photography that leverages imaging devices, signal processing and machine learned models to perform non-contact recovery of vital processes inside the body [23]. Data plays an important role in both training and evaluating these models. However, generalization can be weak if the training data are not representative and systematic evaluation can be challenging if testing data do not contain the variations and diversity necessary. Public datasets (e.g., [55, 28, 4]) have contributed significantly to the understanding of algorithmic performance in this domain. These datasets are time consuming to collect, contain highly personally identifiable and sensitive biometrics (including facial videos and physiological waveforms). It is difficult to collect datasets that contain a well distributed set of examples across multiple cardiac and pulmonary parameters (e.g., heart and breathing rates and variabilities, pulse arrival times, waveform morphologies). Furthermore, almost all of these datasets are collected in a single location, with limited diversity in subject appearance, ambient illumination, context and behaviors. Table 1 summarizes some of the properties of these datasets, including whether they are freely (i.e., at no cost) available to researchers in both industry and academia. Finally, at the time of writing, neural architectures [5, 20, 54] provide the state-of-the-art performance for camera measurement of physiology. Neural models are "data hungry" and often performance is primarily a function of the availability and quality of the training dataset.

Synthetics have proven valuable in several areas of computer vision, particularly face and body analyses. In training, synthetics have been used successfully to create models for landmark localization and face parsing [52], body pose estimation [35] and eye tracking [53]. Although not completely representative of real observations, synthetics are also valuable in testing (e.g., for face detection [24] or eye tracking [40]). Parameterized computer graphics simulators are one way of testing vision models [46, 47, 48, 45, 33]. Generally, it has been proposed that graphics models be used for performance evaluation [13, 33, 24]. However, increasingly synthetics are also being used to help address shortcomings in performance, such as biases. Kortylewaski et al. [17, 18] show that the damage of real-world dataset biases on facial recognition systems can be partially addressed by pre-training on synthetic data. To address the issue of the lack of representation of skin type in camera physiology datasets computational techniques have been employed to translate real videos from light-skin subjects to dark-skin subjects while being careful to preserve the cardiac signals [1]. A neural generator was used in that work to simulate changes in melanin, or skin tone. However, this approach does not simulate other changes in appearance that might also be correlated with skin type. Nowara et al. [30] used video magnification for augmenting the spatial appearance of videos and the temporal magnitude of changes in pixels. These augmentations help in the learning process, ultimately leading to the model learning better representations.

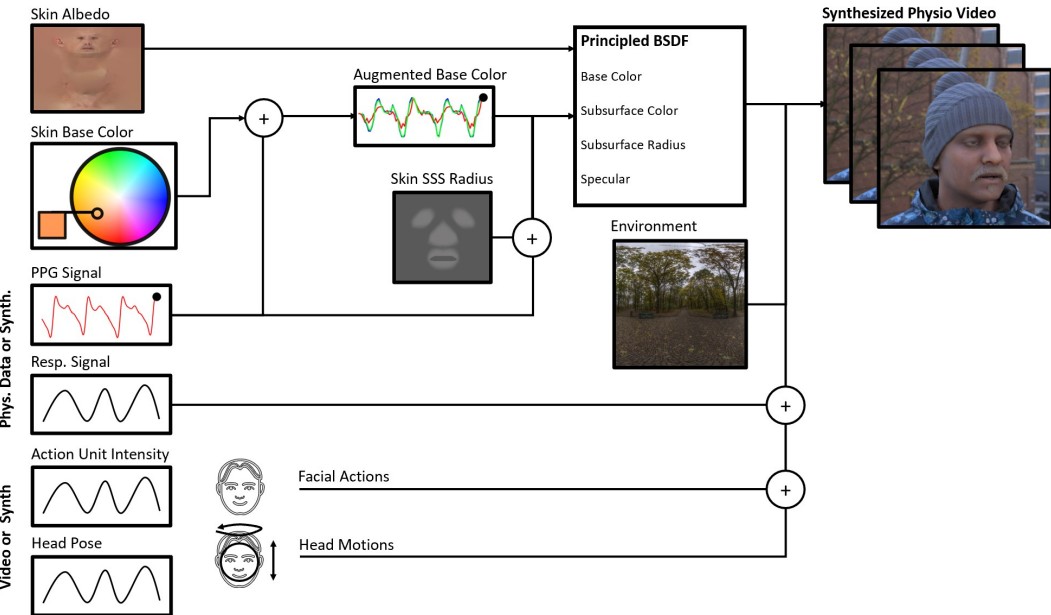

Figure 2: The synthetic videos were created using a graphics pipeline. We create a model of facial blood flow by adjusting properties of the physically-based shading material used for the skin and a model for breathing by controlling the motion of the head and torso. Facial actions and head motions are added to create realism and variability.

Wood et al. [52] recently presented a sophisticated facial synthetics pipeline that produced high-fidelity data. They were able to successfully train state-of-the-art landmark localization and face parsing models. However, creating high fidelity 3D assets for simulating many different facial appearances (e.g., bone structures, facial attributes, skin tones etc.) is time consuming and expensive. The data that these pipelines can create will then not necessarily be available broadly to researchers. Therefore, in this paper we present a new dataset (SCAMPS) of high fidelity synthetic human simulations that are made publicly available. These data are designed for the purposes of training and testing camera physiological measurement methods. To summarize our contributions: 1) We present the first public synthetic dataset for camera physiological measurement. 2) These data include precisely synchronized multi-parameter physiological ground-truth waveforms (cardiac, breathing) alongside facial action and head pose. 3) Results illustrating baseline performance training on the SCAMPS dataset and testing on three public datasets (UBFC-rPPG [4], MMSE-HR [55] and PURE [39]). We hope that this dataset allows researchers to explore the potential for synthetics in the domain of camera physiological measurement, including but not limited to: addressing the simulation-to-real (sim2real) generalization gap, leveraging very precisely aligned segmentation maps and physiological waveforms for learning models, multimodal learning combining estimation of physiological (e.g., HR) and behavioral (e.g., AUs) signals, and using synthetic data to help address bias in camera physiological measurement models.

## 2   Camera Physiological Measurement

Camera measurement of physiological signals involves analysis of subtle changes in light reflected from the body. In videos, the photoplethysmographic signal manifests as small skin pixel color changes over time. The breathing signal is observed as motion, particularly prominent around the chest. Blazek, Wu and Hoelscher [3] proposed the first imaging system for measuring cardiac signals. This computer-based CCD near-infrared (NIR) imaging system provided evidence that peripheral blood volume could be measured without contact using an imager. Successful replications of these experiments cemented the concept [51, 41, 49]. Applying machine learning tools and knowledge of physiological principals helped to create more robust measurement methods [32, 50, 12]. With supervised methods, data soon becomes a limiting factor [38, 5, 37, 22, 54]. The significance of training data is increasing as large parameter models illustrate the potential for representation

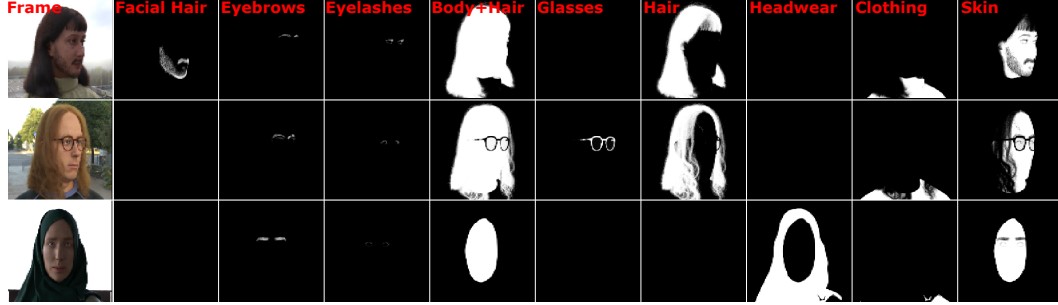

Figure 3: Each RGB frame is accompanied by segmentation masks for facial and body hair, eye-lashes, eyebrows, glasses, skin, head wear and clothing.

learning [54]. Work on body motion analysis from video, has found that to be a rich source of physiological information. enabling the recovery of breathing [42] and cardiac signals [2]. These methods do not require light to penetrate the skin but rather use optical flow and other motion tracking methods to measure, usually very small, motions. These subtle changes are easily swamped by larger body motions and facial expressions. Therefore, an algorithm needs to learn to successfully separate the sources from pixel changes both spatially and temporally [6]. If we subscribe to the results of recent machine learning research, it is likely that supervised models can learn to separate signals more effectively than handcrafted rules. For more comprehensive overviews of video physiological measurement see Chen et al. [7], Shao et al. [34] and McDuff [23].

## 3    Waveform Synthesis

Our synthesis pipeline starts with a module for generating the underlying physiologic and behavioral signals. These signals are then used to drive those properties of the synthetic humans providing precisely synchronized ground-truth labels.[1] Examples of the generated waveforms can be found in Fig. 4. To create physiological waveforms with variability we sampled several waveform parameters, such as heart rate variability standard deviation of NN intervals (HRV SDNN), relative amplitude of the systolic and dicrotic waves and the delay between the systolic and dicrotic waves from a set of uniform distributions. The bounds used for each of these parameters are specified below.

**Inter-beat Interval, PPG, ECG Waveforms.** The PPG and ECG signals were created to have the same underlying beat sequence. We first sample the beat sequence based on a heart rate (HR) frequency sampled uniformly from 40 to 150 beats/min. Heart rate variability is simulated by adding random perturbations to the beat timings. The standard deviation of these perturbations reflects the standard deviation of NN intervals (SDNN) and was sampled uniformly from 0.05 seconds to 8/HR seconds. We observed that it was important for the upper bound to be proportional to the heart rate (or mean NN interval) to create realistic variability.

For the purposes of this simulation, the morphology of the ECG wave is not relevant (e.g., we do not try to simulate a realistic QRS complex), only the timing. Thus, the ECG waveform is constructed as a time delayed series of impulses based on the NN intervals. We provide the interbeat intervals directly so that no peak detection is required for the ground-truth waveforms.

Given the beat timings and pulse arrival time (PAT) the PPG wave was then composed of a forward wave and dicrotic wave. The forward wave is created by convolving a Gaussian window with the beat impulse sequence. The leading slope of the dicrotic wave is created by convolving a Gaussian with a time lagged copy of the beat impulse sequence, the trailing slope is generated by performing the same convolution with a decaying exponential in place of the Gaussian window.

These waves are then summed together with a dicrotic amplitude factor. The forward and dicrotic waves are then superimposed, with parameterized attenuation of the dicrotic wave relative to the forward wave, to create a physiologically plausible PPG waveform.

---

[1]It is important to note that the purpose of our waveform synthesis approach was not to create signals derived from a true physical model of arterial hemodynamics and tissue perfusion, but instead to develop a simple and efficient way to generate physiologically plausible waveforms.

This signal was then low pass filtered to clean up the edges of the Gaussians, using a filter cut-off frequency of 8 Hz. Finally, the signal was normalized to give a signal of maximum amplitude of 1. This process creates PPG waveforms with the characteristic profile of systolic peaks and smaller diastolic peaks or inflections, but also with variability in the form. Finally, a small baseline drift at the breathing frequency is applied to the PPG signal to capture the subtle variations observed with breathing.

**Breathing Waveforms.** Each breathing waveform was created using sequence of breathing times based on a breathing frequency sampled from 8 to 24 breaths/min. A Gaussian window was convolved with the resulting impulse sequence. This signal was then low pass filtered to clean up the edges of the Gaussians, using a filter cut-off frequency of 8 Hz. Finally, the signal was normalized to give a signal of maximum amplitude of 1.

**Facial Actions, Blinking and Head Pose.** Unlike the physiologic waveforms, facial actions (with the exception of perhaps blinking) are rarely periodic. Therefore, we adopt an event based model [44]. For each facial action the event signal was created by a set of ramped step functions. The minimum and maximum event durations were 1 and 4 seconds, respectively. Blinking was treated separately from the other facial actions as the behavior is relatively more frequent and repetitive. For blinks the min and max event durations were 0.3 and 1 second respectively.

In each video we generate action unit "events". The start time and duration since previous event govern when the events onset and the gap between two events of the same action unit. These were are sampled from uniform distributions with bounds [0.3, 18] seconds and [1, 18] seconds, respectively. As such, in videos with action unit events there are examples of the onset and offset of most actions, some multiple times. Because facial actions are sparse but blinking occurs frequently, we generated all videos with blinking (eyes closed) events but only a subset of videos with facial actions, more details are provided below.

## 4 Video Synthesis

**Identity.** To create the avatars a generative 3D face model captures how face shape vary, and change during facial expressions. A blendshape-based rig is used with 7,667 vertices and 7,414 polygons and the identity basis is learned from a set of high-quality facial scans. In the creation of each avatar, we use a texture map transferred from one of the high-quality 3D facial scan as the albedo of the material for creating each face. These texture maps are sampled from a set of 511 facial scans of subjects including a range of skin types/tones, genders and ages. The distribution of gender, age and ethnicity of the subjects who provided the facial scans can be found in [52] (see Fig. 4). While these scans are not uniformly distributed across all demographic profiles, they do provide a wide range of appearances. Ongoing efforts are focused on creating more balanced facial scan dataset to help create even more diverse renderings. As only varying the blood flow signal in the skin is important for our use case the facial hair is removed from these textures by an artist. Then the skin properties can be easily manipulated. Hair (and clothing) are added back in later to create the final appearance.

We simulate blood flow by adjusting properties of the physically-based shading material[2]. We use a similar approach to that described by Wood et al. [52] and McDuff et al. [25]. We want the renders to display both diffuse and specular reflection effects, the diffuse reflection is handled as described below when we simulate blood flow and the specular reflection is controlled with an artist-created roughness map. Specular reflections make some parts of the face (e.g. the lips) shinier than others.

Hair, clothing and other apparel are added back in once the blood perfusion signal has been created. Hair is modeled as over 100,000 individual 3D strands to create a realstic effect. Hair as with clothing then occlude perfusion changes, as would be the case in real life.

**Photoplethysmography.** Changes in diffuse reflection due to blood flow are achieved by varying the surface color and subsurface scattering of the skin texture map. We simulate blood flow by adjusting properties of the physically-based shading material we use for the face. The synthesized PPG waveform is used to drive the temporal changes. We manipulate skin tone changes using the subsurface color parameters. The weights for this are derived from the absorption spectrum of hemoglobin and typical frequency bands from an exemplar digital camera[3] (Red: 550-700 nm, Green: 400-650 nm,

---

[2]https://www.blender.org/
[3]https://www.bnl.gov/atf/docs/scout-g_users_manual.pdf

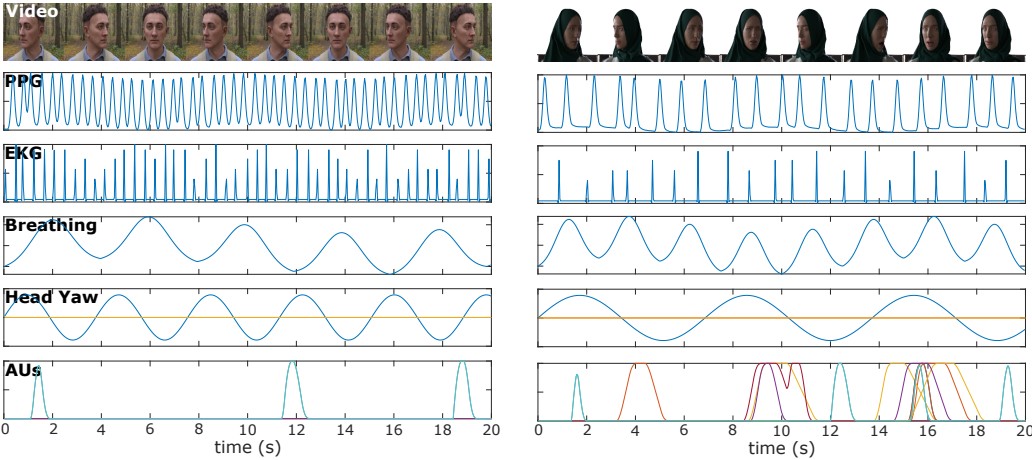

Figure 4: Our synthetic videos are accompanied by frame-level PPG, pseudo ECG/interbeat intervals, breathing, head pose and action unit labels. Here we show examples of two videos with a subset of video frames for reference.

Blue: 350-550 nm). We manipulate the subsurface radius for the channels to capture the changes in scattering as the blood volume varies within the skin. A subsurface scattering radius texture is used to spatially-weight these and simulate variations in the thickness of the skin across the face using an artist-created subsurface scattering radius texture. The same relative weighting of the RGB channels (0.36, 0.41, 0.23) is used for the BSDF subsurface radii. In absence of a more complex temporal-spatial model, we vary the parameters across the skin pixels in the same way across all frames. We recognize this is unlikely to be optimal, but does limit blood flow changes to skin pixels. We hope to be able to introduce a more realistic spatial variation in future. We used relative subsurface scattering coefficients of 0.36 (+/- 0.1), 0.41 (+/- 0.1) and 0.23 (+/- 0.1) for the red, green and blue channels respectively. Empirically we have found that this procedure works for creating data for training camera-based vital sign measurement. We found that varying the subsurface scattering alone, without changes in subsurface color, was too subtle and could not recreate the effects of BVP on reflected light observed in real videos.

**Breathing.** Inhaling and exhaling cause motions of the head and chest. To capture this in the avatars we use an approximation by controlling pitch of the chest and head using the synthesized breathing input signal. The amplitude of the head and chest motions were subtle and when combined with the head rotations and facial expressions are often difficult to see; however, prior validation has shown the models trained on similar synthesized data can generalise to real videos.

**Facial Actions** Facial expressions are controlled using blendshapes that map approximately to 10 facial action units [10]: outer brow raise (AU2), brow lowerer (AU4), eye lid tightener (AU7), lip corner puller (AU12), lip corner depressor (AU15), chin raiser (AU17), lip puckerer (AU18), jaw drop (AU26), mouth stretch (AU27) and eyes closed (AU43). The facial action coding system is a widely used and relatively objective method for quantifying facial movements [10]. The goal of controlling these actions is to create upper and lower facial motions. We recognize that the behaviors do not necessarily simulate realistic talking or expressions, as the dynamics of these are difficult to simulate.

## 5 Dataset

We created a dataset of 2,800 video sequences. The rendering required 24 machines each with an NVIDIA M40 GPU running for 720 hours each (a total of 17,280). This illustrates that creating synthetic data of this kind is not trivial and in part justifies the need for public datasets that can be shared amongst researchers. Each video has frame level ground-truth labels for PPG, inter-beat (RR) intervals, breathing waveform, breathing intervals and 10 facial actions. We also provide video level ground-truth labels for HRV SDNN, r-peak pulse arrival time (rPAT) and dicrotic wave amplitude. These parameters were used to generate a set of 20 second PPG waveforms at 300Hz. Finally, action

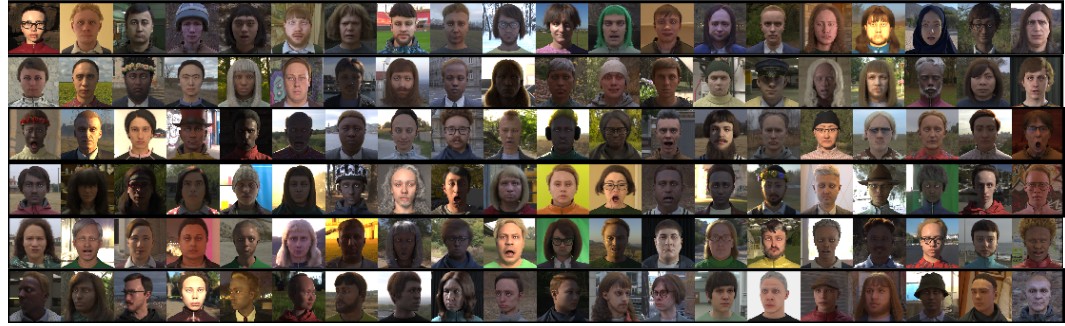

Figure 5: Example frames from the SCAMPS dataset showing the diversity in avatar appearance, behavior and environment.

unit intensities were generated. The ground-truth metrics are provided as both MAT and CSV files. Each video was then rendered using the corresponding waveforms and action unit intensities, and randomly sampled appearance properties, including skin texture, hair, clothing and environment.

Figure 6 shows the distribution of heart rates, HRV SDNNs, dicrotic wave amplitudes and breathing rates in the dataset. HR, rPAT and dicrotic wave amplitudes were sampled uniformly. HRV SDNN was not sampled uniformly, as qualitatively large HRV values, while interesting, could create quite extreme differences in interbeat intervals and we deemed it appropriate to create more examples with smaller variability.

To create a dataset that can be used for training and testing under a diverse range of conditions we synthesized videos while systematically changing different confounders: 1) head motions, 2) facial actions, and 3) dynamic illumination. A training, validation and test split of the data is provided on our project page as is a file indicating which confounders are present in each video. As each video was sampled with a different combination of appearance parameters, they all contain avatars with different appearance. However, some avatars may look similar if they have the same skin texture and hair style. Figure 1 and 5 both show a collage of frames from different videos illustrating the diversity in appearance. The video frame (RGB) come with segmentation maps (see Fig. 3) that provide pixel level labels for beard, eyelashes, eyebrows, glasses, hair, skin and clothing. This is important as we know that the PPG signal will not be present in material that do not have blood flow (e.g., hair, clothing) and so we expect any supervised learning method to learn to segment skin as one of the operations. Therefore, we anticipate that segmentation maps will be useful to the community, both in training and in testing camera PPG methods.

**Head Motions.** Two thousand videos have rotation head motions and 800 have no head motion. Of the videos with head rotations, 1200 have smooth rotation (400 videos at 10, 20 and 30 degrees per second) and a further 800 have non-smooth head rotations in which the head was randomly positioned every second to a different angle. Ground-truth head angles are provided in the label files.

**Facial Actions.** Half of the videos (1,400) have facial actions generated with the event model described above, the other half have no facial actions. This enables training and/or testing systematically introducing the confounder of facial motions on the physiological measurement. The sequences and combinations of facial actions in each video were randomly sampled and therefore some of the facial expressions can look unnatural; however, this does provide a relatively dense set of examples of facial action onsets and offsets. We contrast this to many facial expression datasets in which facial actions are relatively sparse. We felt that more examples would generally be more useful for training models.

**Background Motion and Dynamic Illumination.** A set of 400 of the videos have dynamic illumination and background motion created by simulating the subject turning around in the environment. Half of these 400 videos have facial actions and half have head motions in addition to the background motion.

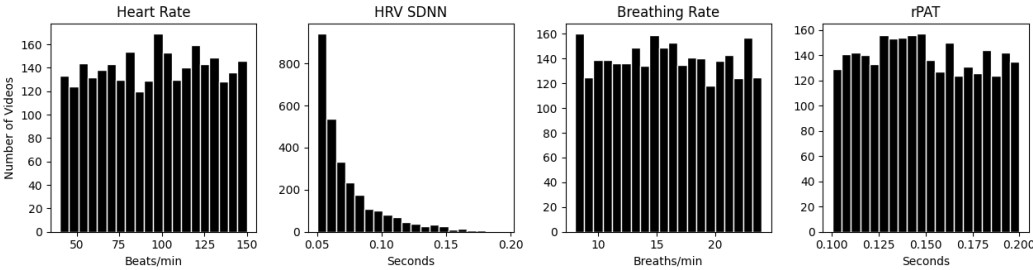

Figure 6: Examples of the distribution of heart rates, HRV SDNNs, breathing rates and dicrotic wave amplitudes in the SCAMPS dataset. An advantage of synthetic data pipelines is the ability to create a wide range of examples with specific distributions.

## 6  Baselines

One might ask the question "how well does a model trained on synthetic data generalize to real videos?" While there is some precedent for using synthetics for heart and breathing rate estimation [25, 26], those works did not use the SCAMPS dataset. To illustrate how this specific dataset can be used for video physiological measurement and provide initial baseline results, we performed experiments training with the SCAMPS dataset and testing on two public benchmark video datasets. To generate the results in this paper we used the opensource Deep Physiological Sensing Toolbox [21]. Links to the trained models can be found on our project page.

**Model.** Our goal here is not to provide an exhaustive list of results on different model architectures, but a representative baseline for researchers to compare to. We do not argue that this is the current state-of-the-art but rather is a reasonable starting point for future research with synthetic data in the field of camera physiological measurement. We implemented DeepPhys [5] as the baseline supervised model due to its relative simplicity. We trained on frames with resolution 72x72 pixels. First, we cropped the center 240x240 pixel region of each 320x240 pixel raw images. We then down sample these to 72x72 using a bilinear downsampling method. Difference frames were computed by performing a difference operation on successive frames. The resulting appearance and difference frames were normalized consistent with the method in Chen and McDuff [5]. These frames are then used for training the supervised model. We used a learning rate of 0.001 and the ADAM optimizer. We trained the model using videos from the SCAMPS training set for 10 epochs. We validated the SCAMPS validation set but used real-world videos as the testing sets. We used the Deep Physiological Sensing Toolbox [21] to complete all the training and testing procedures. The model from the epoch with lowest mean absolute error (MAE) heart rate estimation was selected and then we evaluated this model on the test sets. A Butterworth filter was applied to all model outputs (cut-off frequencies of 0.7 and 2.5 Hz) before computing the frequency spectra and heart rate.

**Results.** The results reported here are on the UBFC-rPPG [4], MMSE-HR [55] and PURE [39] datasets. Table 2 shows the mean absolute error (MAE), root mean squared error (RMSE) and correlation ($\rho$) in heart rate estimation compared to the gold-standard measures from each of the datasets. The results on both datasets show that the synthetic data are sufficient to train a reasonable supervised model. The trained model does not necessarily exceed the performance of the existing unsupervised methods and is in some cases a little worse. However, as first baselines these numbers do demonstrate that generalization from synthetic video to real ones is possible and also that there is room for improvement. By releasing the SCAMPS dataset we hope that researchers can design methods that bridge the sim-to-real gap that exists.

## 7  Access and Usage

The data may be used for research purposes and any images from the dataset can be used in academic publications. Researchers may redistribute the SCAMPS dataset, so long as they include all credit or attribution information and that the terms of redistribution require any recipient to do the same. The license agreement details the permissible use of the data and the appropriate citation, it can be found at: https://github.com/danmcduff/scampsdataset. Use of the dataset for

Table 2: Cross-dataset heart rate evaluation on UBFC, MMSE-HR and PURE (beats per minute).

| Method | UBFC [4] | | | MMSE-HR [55] | | | PURE [39] | | |
|---|---|---|---|---|---|---|---|---|---|
| | MAE↓ | RMSE↓ | ρ↑ | MAE↓ | RMSE↓ | ρ↑ | MAE↓ | RMSE↓ | ρ↑ |
| DeepPhys[5] (trained on SCAMPS) | 3.74 | 12.42 | 0.82 | 4.59 | 8.89 | 0.81 | 3.51 | 12.94 | 0.84 |
| POS[50] | 3.52 | 8.38 | 0.90 | 3.90 | 9.61 | 0.78 | 1.68 | 9.60 | 0.92 |
| CHROM[9] | 3.10 | 6.84 | 0.93 | 3.74 | 8.11 | 0.82 | 6.23 | 17.18 | 0.71 |
| ICA[32] | 4.39 | 11.60 | 0.82 | 5.44 | 12.0 | 0.66 | 5.70 | 18.10 | 0.70 |

MAE = Mean Absolute Error in HR estimation (Beats/Min), RMSE = Root Mean Square Error in HR estimation (Beats/Min), ρ = Pearson Correlation in HR estimation.

commercial purposes is strictly prohibited, although research use at commercial companies is permissible. The authors commit to maintaining the dataset and ensuring access is available to the research community.

Some of our rendered faces may be close in appearance to the faces of real people. Any such similarity is naturally unintentional, as it would be in a dataset of real images, where people may appear similar to others unknown to them. As such there is no personally identifiable data or biometrics contained within the data, but the authors bear responsibility in case of any violation of rights that might occur.

## 8   Transparency and Broader Impacts

This dataset was created for research and experimentation on camera measurement of physiological signals. While the dataset is useful for testing models, was not designed as a test set for evaluating the clinical efficacy of a model, just because a model performs well on synthetic data does not mean it will generalize to videos of real people. The SCAMPS dataset was not designed for computer vision tasks such as face recognition, gender recognition, facial attribute recognition, or emotion recognition. We do not believe this dataset would be suitable for these applications without further validation.

We have tried to make this dataset representative of a diverse population and the physiological waveforms are completely synthesized, so do not contain identifying information. However, our dataset still does not capture a uniform distribution of skin types and other appearance characteristics. We are working on addressing these limitations. When using this dataset, as with others, one should be careful to pay attention to biases that might exist. Please see the SCAMPS dataset datasheet [11] included in the supplementary material and linked from our project page for more details.

Non-contact camera-based vital sign monitoring has great potential as a tool for telehealth. Our proposed system can promote global health equity and make healthcare more accessible for those in rural areas or those who find it difficult to travel to clinics and hospitals in-person (perhaps because of age, mobility issues or care responsibilities). These needs are likely to be particularly acute in low-resource settings. An advantage of camera physiological measurement is that contact with the body is not required and that cameras are ubiquitous sensors. However, these advantages can cause problems. Unobtrusive measurement from small, ubiquitous sensors makes measurement without a subject's knowledge simpler. It is important that norms and regulations that govern on-body physiological measurement devices are extended to camera measurement systems. Consent should always be obtained from subjects before measuring physiologic data of this kind. It is always important to consider how such technology could be used by "bad actors". In the case of physiological measurement, it should be required to inform subjects when these methods are being used and for consent to be obtained before physiological data is measured or recorded. There should be no penalty for individuals who decline to be measured.

## 9   Future Directions

The SCAMPS datasets is a first of its kind. Therefore, we wanted to only include renderings for which we had a sufficiently robust synthetics pipeline. In the SCAMPS dataset we did not synthesize videos with very abnormal rhythms, or specific types of arrhythmia (e.g., Premature Ventricular Contraction - PVC, Atrial Fibrillation - AFib., etc.) A distinct advantage of synthetic data generation

is the ability to create examples of rare events "at will"; however, creating data that are faithful to real-world observations is non-trivial. Therefore, the first version of the SCAMPS dataset contains pulse signals with heart rate variability, but not specific arrhythmia. We hope that future research will address this gap.

To make the dataset more plausible, a simulation of ballistic forces (e.g., ballistocardiogram) would be helpful, as would a more sophisticated absorption model that reflects how absorption might change under different conditions. Simulating scar tissue, makeup and other skin markings (e.g., tattoos or piercings) would also help provide better representation of appearances to the dataset. Our current rendering engine is not capable of simulating scar tissue and the skin albedos we used did not have tattoos. These are examples of why it is important to pay attention to biases that might exist in models trained with SCAMPS and why it would not be appropriate to deploy a model trained on SCAMPS without further work.

## 10    Conclusions

The SCAMPS dataset contains high-fidelity simulations designed for training and testing camera-based physiological sensing algorithms. The dataset was designed to capture a diverse range of appearances, environments and lighting conditions. Synchronized ground-truth signals include interbeat and breath intervals and PPG, ECG and breathing waveforms precisely aligned with each video frame. Facial actions, blinking and head pose labels are also provided. Benchmark experiments show that it is possible to train models only with these synthetic data that generalize to real videos. We hope that this dataset helps support research towards more robust and fair vision-based physiological sensing models.

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
