# OpenReview forum: "SCAMPS: Synthetics for Camera Measurement of Physiological Signals"
_NeurIPS.cc/2022/Track/Datasets_and_Benchmarks — NeurIPS 2022 Datasets and Benchmarks _

### Official Review · Reviewer_BKBx · 2022-07-20
**Very good approach at generating synthetic physiological data.**

**Rating:** 8
**Confidence:** 4

**Strengths:**

The proposed dataset claims that no personal information is shared.

Moreover, the dataset includes occlusion masks that can help experimental segmentation designs.

It is worth noting that the authors illustrate generalizability on a real-world public dataset in their experiments.

**Weaknesses:**

Despite the clear contributions of this dataset towards new methods in camera measurement of physiological signals, the Authors should discuss some issues further.

One of such discussion points should be simulating scar tissue and its potential impact on heart rate measurement.

Moreover, makeup or other skin augmentation such as tattoos or piercings could be included and mentioned explicitly.

**Additional Feedback:**

Questions towards this work would be as follows:
- Can you provide the code that was used for baseline experiments.
- Will the infrastructure for generating this dataset be published as open-source?
- Do you plan on including skin augmentation and makeup in the future version of this dataset.

**Clarity:**

Although most of the article is correct, the authors could address minor clarity issues. Unfortunately, the authors provided a version of the article that does not contain line numbers, so it makes pointing them out a little challenging nevertheless, please refer to the issues below:
- The "Introduction" section contains a minor stylistic issue, where the word "physiological" is used twice in the same sentence.
- Table 1 contains datasets that were previously available. When showcasing the available variables, two are similar enough that they might be discussed in more detail. This is especially relevant in the case of the MAHNOB dataset containing "Breath - Breathing waveform" and "BR - Breathing rate". It seems possible to calculate breathing rate from the breathing waveform. This could be discussed further.
- Section 4 "Video Synthesis", paragraph "Facial actions" seems to contain a typo: "lid tightened" instead of "lip tightened". Moreover, the same paragraph mentions that the system used for action coding is widely used. There seems to be a lack of citation towards this system or its full explicit definition. Pointing readers toward this system documentation might be a good step toward extending the clarity of this work.

Within the article, there seem to be a few instances where past tense could be used, but it is a matter of stylistic preference rather than the merits of the introduced dataset. On the other hand, I would recommend reviewing the whole article once again and verifying if the authors think the writing could be improved. In some instances, "will be publicly available" should be "are publicly available".

**Correctness:**

The claims presented in this paper are not exaggerated. Authors build upon some existing work and provide valuable material.

Unfortunately, the code that the Authors used for experiments was not published.

**Documentation:**

The official repository contains a more precise explanation of the data format, including smaller samples for preview.

The authors included a datasheet for the dataset required for introducing new datasets.

The license for data is clearly stated. On the other hand, there is no maintenance or hosting plan information. The dataset itself is not hosted in a persistent scientific repository.

**Ethics:**

The authors present any potential for dataset misuse.

The only ethical issue with this article would be that face scans were used and briefly mentioned. These could be discussed a little bit more within the body of this work.

**Relation To Prior Work:**

Authors claim their dataset is a first attempt at generating synthetic data for camera measurement of physiological signals - verifying that information is impossible. A straightforward comparison is made to other camera measurement physiological datasets.

**Summary And Contributions:**

The authors proposed a synthetic dataset that could lower the costs of collecting physiological data using camera measurement methods.

This dataset is especially relevant as, in real-world scenarios, the synchronization of multimodal data sources is a challenge.

Moreover, the authors own the infrastructure for the synthesis of this dataset. Future updates are a possibility.

---

> ### Author Response · Authors · 2022-08-12
> **Response to Reviewer BKBx**
>
> We would like to thank you for your positive and very constructive review.  Your suggestions are discussed below touch on valuable topics, some of which we had not discussed in the original version of the paper.
>
> Simulating scar tissue, makeup and other skin appearances (e.g., tattoos or piercings) is a very interesting suggestion. Our current rendering engine is not capable of capturing simulating scar tissue and the skin textures did not have tattoos.  We have added words to this effect in the Future Directions section of the paper.  We believe that such capabilities would be very valuable and increase the diversity of the data that we could create.
>
> We would like to clarify that consent was obtained for the facial scans to be used to create avatars and were licensed under that understanding.
>
> In the Broader Impacts statement we included some discussions of negative applications of these technologies in surveillance and use by “bad actors”.  But we appreciate your concerns have expanded this discussion in the latest version. We think it is of fundamental importance that consent is obtained for any type of physiological measurement and that there are clear mechanisms to opt out.  Scalable digital health measurement technologies have utility and could provide significant benefits, but not at the cost of surveillance.
>
> We appreciate the comments on the stylistic issues in the paper.  We have addressed these issues in the revision.  We apologize for not providing a version with the line numbers.  We have corrected that in the revised version.

---

### Official Review · Reviewer_vXzY · 2022-07-22
**A novel synthetic dataset for physiological signals estimation from RGB videos**

**Rating:** 7
**Confidence:** 3
**Clarity:** The paper is clearly written.

**Strengths:**

The proposed a novel synthetic dataset is diverse in terms of populations, environments and confounding factors. The dataset is large, having 2800 RGB videos, totalling 1.68M frames.

The authors took care into providing simulated semi-realistic physiological patterns in terms of PPG, pulse rate and breathing, which are synchronized with the videos. Benchmark results showcasing direct transfer to real-world datasets imply a promising direction in the field.

**Weaknesses:**

The authors simulate heart-rate variability by applying random perturbations to the beat timings. However, there is also a relationship between breathing and heart rate, in the form of respiratory sinus arrhythmia, in which the heart rate synchronizes with the breathing. Moreover, the heart rate increases slightly on the inhale, and slows down on the exhale, which I’m assuming affects photoplethysmography readings in some subtle ways.

It was not clear to me if the heart rate and breathing patterns are linked in the simulated videos. If not, why?

The videos have a heavy focus on side-to-side rotations of the head. Why isn't there more variability with regards to the camera angle (not only horizontal but also vertical angles)?

**Additional Feedback:**

Overall, it is a good contribution, being the first synthetic dataset designed for physiological signs estimation.

**Correctness:**

The dataset is constructed in a sound way. Benchmarks do not have explicit accompanying code, but the authors indicate an external tool to reproduce the results.

**Documentation:**

The dataset has comprehensive documentation on the project github page, supplementary material and main paper.

**Ethics:**

While the dataset itself has no ethical issues, its application might. The authors claim that "Unobtrusive measurement from small, ubiquitous sensors makes measurement without a subject’s knowledge simpler", which indicates that such models might be used for ubiquitous surveillance. I think further discussion is needed in the paper, from an ethical point of view, on the potential applications of estimating physiological from camera sensors.

**Relation To Prior Work:**

The authors clearly discussed in the Introduction the differentiating aspects between their proposed synthetic dataset and existing real-world datasets.

**Summary And Contributions:**

The works presents SCAMPS, a synthetic dataset for estimating physiological signals (PPG, pulse, breath) from human avatars. The authors provide 2800 high quality videos with a diverse range of simulated human faces and environments. The authors introduce facial action units and multiple types of head movements (side-to-side, up and down), which are influenced by breathing.

---

> ### Author Response · Authors · 2022-08-12
> **Response to Reviewer vXzY**
>
> Thank you for your review and for your encouraging and insightful comments.
>
> The PPG and breathing signals are linked in the simulated videos to some degree. We intentionally synthesize the waveforms with a correlation between the respiratory waveform and the low frequency components of the PPG waveform. However, we recognize that the relationship is quite simplistic and does not perfectly model respiratory sinus arrhythmia (RSA). A more elaborate model would be desirable; however, we do think that the SCAMPS dataset is still an effective training resource for remote PPG research without it.  We emphasize that more work needs to be done before training medical devices on these data, and the imperfect RSA modeling is an example of why that might be the case.
>
> We deemed side-to-side (yaw) rotations of the head to be more common than pitch rotations in many scenarios (e.g., telehealth video calls). We could not create all possible forms of head motion and translation in this dataset and the combinations are just too large due and the computational cost involved in generating the videos would be too great. Therefore, we prioritized systematic yaw rotation, facial expressions and lighting changes as the main properties.
>
> We appreciate you highlighting the ethical concerns. We believe that tools for scalable digital health measurement are very valuable; however we also agree that any form of physiological sensing should only be performed with consent from the subject and that consent should be comprehensible and clear, camera sensing is no exception.  There are many ways to measure physiological signals, using motion sensors, WiFi, cameras, audio signals and so we need mechanisms to obtain consent before ubiquitous sensing such as this is deployed.  We have extended the Broader Impacts section and agree that this is a very important discussion.

---

### Official Review · Reviewer_7ZvN · 2022-07-27
**Diverse synthetic multi-modal dataset to measure physiological signals**

**Rating:** 8
**Confidence:** 4
**Correctness:** Yes.
**Clarity:** Yes.

**Strengths:**

1. Valuable contribution: A multi-modal dataset which is perfectly synchronize to measure physiological signals (using only videos) is challenging to capture. Therefore, paper's contribution of a large synthetic dataset in this field is valuable for the community. In addition to training models to measure physiological signals, this dataset can be used to train/pre-train segmentation models, multi-modal self-supervised models, conditional GANs, etc.
2. Non-trivial synthetic dataset: Creation of this dataset is non-trivial. To create this dataset, it requires diverse know-how of the interactions between lighting conditions, environmental conditions, skin tones and physiological signals. Additionally, the computational resources required for creating this dataset is also not easily available (24 machines each with an NVIDIA M40 GPU running for 720 hours each)
3. Diverse dataset: The dataset of 2800 videos (1.68 million frames) covers a range of environments, body motions, illumination conditions, and physiological states.
4. Overall paper quality:  The paper is well written. It anticipates questions of the reviewers and addresses them in the manuscript.


**Weaknesses:**

1. Although the entire pipeline for generating the synthetic dataset is explained in the paper, it would not be enough to reproduce it. Would it be possible for the authors to share a detailed code/ step-by-step guide for generating the dataset? This can/may include steps that require external tools/library like Blender.org.
2. To alleviate any ethical concerns with respect to gender, race or age biases, it would be nice if the authors can include their distributions in the appendix (separate distribution for gender, race and age should work)
3. Since there are some real world datasets, it would interesting to see if a model pre-trained on this synthetic dataset and then finetuned on the real world data is better than a model just trained on the real-world data. This would further emphasize the utility of the dataset if models pretrained on synthetic dataset out-performs other pre-trained/ trained from scratch models.

**Additional Feedback:**

None

**Documentation:**

Reproducibility of generating the synthetic datasets can be improved. Please see point 1 in the weaknesses.

**Ethics:**

There are some potential ethical concerns. They also discussed in the paper.
1. Biases against gender, race and age: The dataset could bias towards a certain population. To ensure more transparency, it would be advisable to the authors to include some distribution of the age, race, gender, or their combination in the appendix.
2. Privacy concern: This dataset (and many other datasets for measuring physiological signs using camera) can help to build models to measure personal information like physiological signals in an obtrusive manner where the subject is unaware of being measured. Therefore, I would recommend the authors to make their data open access only after some kind of registration. There are platforms like simtk.org that allows the researchers to host their data which is accessible to the viewers only after having a simtk account.

**Relation To Prior Work:**

Yes.

**Summary And Contributions:**

The paper introduces a synthetic multi-modal dataset that can useful for training models to measure physiological signals using face-captured videos. The authors have used their domain expertise to embed subtle changes in the skin tones of an avatar as breathing and other physiological signs vary. This way one can obtain a perfectly synchronized multi-modal dataset to train the models for measuring physiological signals. The dataset of 2800 videos (1.68 million frames) covers a range of environments, body motions, illumination conditions, and physiological states. The authors have also ensured the dataset is diverse in terms of skin tone, gender and age. Since this is a synthetic dataset, it is also accompanied with segmentation masks for hair, bread, skin, etc. They show that a model trained on the synthetic dataset has decent generalizability on the real-world dataset.

---

> ### Author Response · Authors · 2022-08-12
> **Response to Reviewer 7ZvN**
>
> Thank you for your detailed review and helpful comments. We are encouraged by your comments recognizing the value of this dataset and how it was non-trivial to create.  Given the computation resources required to create such a dataset we wanted to make sure as many researchers as possible had access to it.
>
> We tried to include as much information about the rendering pipeline in the paper but recognized we were short on space.  We have added more information to the paper (in Section 4) and have referenced two other works [1,2] that provide additional details on the rendering pipeline. [1] provides detailed mathematical details about the rendering pipeline, but we felt it was redundant to replicate that in this paper.  We have clarified that this is where the readers can find more details in the revision.
>
> The distribution of gender, age and ethnicity in the facial scans can be found in Wood et al. [1] (Fig.4). It was an oversight not to include that information in the original version of the paper. Thank you for pointing it out.  While this distribution is not uniform across all ethnicities and age, our synthesized videos contain a large variety of appearances, including skin types and textures.  Beyond this there is also a lot of variability in clothing and backgrounds.  The backgrounds are sampled from ​​HDRI Haven that contains 448 backgrounds which were sampled uniformly.  Our dataset contains 2800 videos and therefore each background appears in approximately six different videos.  More information about the HDRI Haven scenes can be found in [2]. We did not quantify the illumination intensity, but this would be a good piece of metadata to add to the dataset.
>
> The suggestion for fine-tuning and/or including real-videos at training time to see if that bridges the sim-to-real gap is a good one. By releasing this dataset we hope that such experiments can be performed by the researchers.  It has already been demonstrated that a mixture of synthetic and real data can help improve the performance of remote PPG measurement [2]. However, there are many other ways synthetic and real data could be combined that are yet to be fully explored.
>
> [1] Wood, E., Baltrušaitis, T., Hewitt, C., Dziadzio, S., Cashman, T. J., & Shotton, J. (2021). Fake it till you make it: face analysis in the wild using synthetic data alone. In Proceedings of the IEEE/CVF international conference on computer vision (pp. 3681-3691).
>
> [2] McDuff, D., Hernandez, J., Liu, X., Wood, E., & Baltrusaitis, T. (2022). Using High-Fidelity Avatars to Advance Camera-based Cardiac Pulse Measurement. IEEE Transactions on Biomedical Engineering.
>
> We appreciate you raising the concerns about potential biases in the dataset.  We believe that this dataset is a step toward more representative data.  However, we do recognize that it is not perfectly balanced across all dimensions.  We have added clearer wording to this effect in the paper in the Video Synthesis section.
>
> Regarding the privacy concerns, we do believe there are very positive applications for scalable digital health sensing that can enable health assessments in more locations and place less demand for financial resources. However, we need to make sure that the data are not used for negative applications. We thank you for suggesting simtk.org as a way for researchers to access the dataset only after they have an account.  We are investigating this option and will post an update here shortly.

---

### Official Review · Reviewer_gcMH · 2022-07-27
**The paper proposes an interesting synthetic dataset of face videos with physiological signals, but would benefit from a better documentation.**

**Rating:** 7
**Confidence:** 3
**Correctness:** Yes.
**Clarity:** Yes. The paper is well written and ea…

**Strengths:**

(+) The paper proposes an interesting synthetic dataset with videos of faces, where physiological signals are present. It is quite unique as it combines rendering techniques from computer graphics with physiological signals to make the human faces more realistic.

(+) The paper evaluates the pertinence of such dataset on existing models and show its utility on existing evaluation datasets for the task of heart rate prediction.

**Weaknesses:**

(-) The dataset lacks evaluation on its physiological plausibleness
* While the paper claims to be physiological plausible by including physiological signals such as the heart rate or breathing movements, there is no evaluation to assess this physiological plausibleness.
* What would be the next steps to make the data more plausible? e.g., would ballistic forces from the aorta be useful?

(-) The dataset lacks details on subject and scene parameters distribution
* While the paper claims to stem from "a range of skin types/tones, genders and ages", it doesn't provide any information on these demographics statistics regarding the original facial scans or the rendered videos in the proposed dataset. Such information matters to understand any potential fairness bias of the dataset or audit potential model issues (e.g., heart rate is usually more challenging to perform on darker skins).
* It would be interesting to also know the distribution of the background scenes used, as well as the illumination intensity. Such information would inform about the diversity of the scene and could help understanding potential model issues (e.g., not working well in night time).

(-) The dataset lacks explanation regarding the rendering process
* It would be appreciated if some parameters choice would be better justified though (e.g., heart rate in Sec 3, or subsurface scattering coefficients in Sec 4).
* On a side note, would the rendering engine be open-source for the community to generate their own dataset? (e.g. it could be useful to understand the causal effect of some parameters by changing them one by one rather than having random parameters for every video).

(-) The limitations and terms of use could be expanded regarding physiological information
* For example, it is possible to infer a person identity from the QRS complex. Such potential usage should be discussed.

(-) No comments on subject informed content
* Given that the dataset stems from facial scan of humans, it is important to know whether they provided their informed consent for the dataset creation.

**Additional Feedback:**

n.a.

**Documentation:**

Yes. The supplementary materials include a datasheet.

**Ethics:**

No.

**Relation To Prior Work:**

Yes. It is appreciated that the paper has trained several related models on the proposed dataset.

**Summary And Contributions:**

The paper proposes a synthetic face-centric video dataset, with physiological- and biologically-plausible properties.
To achieve this, the authors start with facial scans of real subjects, and render synthetic videos with a computer graphics pipeline by including physiological signals (e.g., heart rate) as well as human movements (e.g., breathing movements, face movement...).
While interesting, the dataset would benefit from better insights in the rendering process and distribution information about its properties.

---

> ### Author Response · Authors · 2022-08-12
> **Response to Reviewer gcMH**
>
> We would like to thank you for your review and are glad that you found this dataset to be a unique and useful contribution.  Your comments were very insightful and constructive.
>
> Evaluation to assess the physiological plausibleness:
> Your review is correct, in that our validation of the physiological plausibility of the synthesized data is only based on empirical evidence - i.e., using measurement error of a model trained on SCAMPS and tested on real videos.  By and large, the results are reasonable given the models were trained solely on synthetic data, but there is still a domain gap and we hope that this spurs further research into how this gap might be addressed. We have emphasized this in the results section of the paper.
>
> To make the data more plausible, a simulation of ballistic forces would be helpful, as would a more sophisticated absorption model that reflects how absorption changes under different conditions. These are very good observations. We have added this suggestion to the Future Directions section. Nevertheless, we believe that the dataset as it is does still provide utility to the research community.
>
> The distribution of demographics and background scenes:
> The distribution of gender, age and ethnicity in the facial scans can be found in Wood et al. [1] (Fig.4). It was an oversight not to include this information in the original version of the paper. We have now added it to the Video Synthesis section.  While this distribution is not uniform across all ethnicities and age, our synthesized videos do contain a large variety of appearances, including skin types and textures. We would argue that the demographics and scans and the resulting synthesized avatars are considerably more diverse than the demographics in existing real remote PPG datasets.  Beyond this there is also a lot of variability in clothing, lighting conditions and backgrounds too.  The backgrounds are sampled from ​​HDRI Haven that contains 448 backgrounds which were sampled uniformly.  Our dataset contains 2800 videos and therefore each background appears in approximately six different videos.  More information about the HDRI Haven scenes can be found in [2]. We did not quantify the illumination intensity, but this would be a good piece of metadata to add to the dataset.
>
> [1] Wood, E., Baltrušaitis, T., Hewitt, C., Dziadzio, S., Cashman, T. J., & Shotton, J. (2021). Fake it till you make it: face analysis in the wild using synthetic data alone. In Proceedings of the IEEE/CVF international conference on computer vision (pp. 3681-3691).
>
> [2] https://polyhaven.com/hdris
>
> The choice of heart rates was selected as being a reasonably realistic range of HRs in an adult population. We should note that this range of heart rates [40-150] is as large or larger than the range of heart rates in any existing remote PPG datasets. However, it is possible that heart rates outside this range could be observed. Similarly, for breathing rates we chose the range based on observations of breathing rates in existing datasets and analyses. The rendering engine itself cannot be released at this time for commercial and licensing reasons. We do agree that a publicly available engine would be useful to the community.
>
> It is correct that identifying information can be extracted from real physiological waveforms.  However, the physiological signals in this dataset were completely synthesized using a parametric model and therefore the waveform shapes do not correspond to any particular person. We have clarified this in the paper.
>
> The subjects from which the original facial scans were taken agreed to the use of their data for rendering synthetic avatars and for distribution of these assets. It is also important to emphasize that the appearance of the synthesized avatars are quite unlike the original participants who provided the facial scans as the albedo and skin texture is only one of the assets used to create a render. Facial hair, facial structure, hairstyle and color, eye color, clothing and other apparel can all be different meaning the final renderings are very unlikely to resemble any of the original participants in a meaningful way.

---

> > ### Comment · Reviewer_gcMH · 2022-08-26
> > **Comment to the response**
> >
> > Thank you for addressing my concerns and updating the paper accordingly. Happy to increase my score.
> >
> > I appreciate the improved discussions on the limitations and future directions.
> > One remaining weakness, also raised by other reviewers, is the non-availability of the rendering engine, which limits future users to actually fully benefit the advantages of synthetic data generation.

---

> > > ### Author Response · Authors · 2022-08-28
> > > **Thanks for your comments.**
> > >
> > > Thanks again for your comments and also for increasing your score.  The feedback has been very helpful.
> > >
> > > We appreciate that the rendering engine itself would be a useful resource.  An open version of the rendering engine is something we will work towards in future.
> > >
> > > However, we want to emphasize that we do not think that this reduces the utility of the generated dataset itself.  Even with access to a rendering engine, creating a dataset the size of the one we are releasing, is still computationally demanding and costly.  By releasing the dataset we are hopefully able to save other research groups these resources.
> > >
> > > Thank you again for all your comments.

---

### Official Review · Reviewer_xdpX · 2022-07-28
**Synthetic visual physiological signals dataset**

**Rating:** 7
**Confidence:** 2
**Correctness:** The physiological modelling assumptio…
**Clarity:** Paper is well written.

**Strengths:**

- Supporting scalable digital health is an urgent societal need. And although this early work is not perfect, it is a very worthwhile endeavour
- High-fidelity modelling and data synthesis with open-source Blender
- Good volume of data

**Weaknesses:**

- while initial distribution of supported physiologies in Fig. 6 is very good, a wider range of normal sinus rhythm (e.g., below 50 and above 150) could make the dataset more representative---as will the inclusion of more exotic morphologies (e.g, arrhythmia)
- limited evaluation (cf. Table 2)


**Additional Feedback:**

Extending this initial effort with QRS-complex-/morphology-accurate modelling could make possible future work beyond vision-based estimation, e.g., by using 3D Blender models (pre-rendering) for say ray-tracing wireless responses.
For more information on morphologies to support in the future, see ECG-related literature, and commercial offerings for typically supported physiological ranges in the medical domain:
[1] A. T. Reisner, G. D. Clifford, and R. G. Mark. The physiological basis of the electrocardiogram. Artech House Publishers, 2006.
[2] Rigel Medical. Rigel 333 ECG Patient Simulator. http://www.rigelmedical.com/products/simulators/ ecg- patient- simulator.
[3] Laerdal Medical Limited. SimMan Patient Simulator.
http://www.laerdal.com/gb/doc/86/SimMan.


**Documentation:**

Accompanying code repo and a dataset datasheet are provided.


**Ethics:**

N/A (synthetic data)

**Relation To Prior Work:**

Work is well positioned.

**Summary And Contributions:**

The authors present a synthetic dataset for visual physiological signals estimation. Physiological signals supported include respiration and heart rate markers. Authors demonstrate viable physiological markers estimation on top of their visual synthesis flow.

---

> ### Author Response · Authors · 2022-08-12
> **Response to Reviewer xdpX**
>
> Thank you for your review. We appreciate that you recognize the importance of scalable digital health tools. We also agree that this work is a first step towards synthetics for training remote physiological measurement systems and that there is scope for further improvements/extensions in future.
>
> The distribution of physiologies:
> While there is heart rate variability in the videos in our dataset, we did not synthesize videos with very abnormal rhythms, or specific types of arrhythmia (e.g., Premature Ventricular Contraction - PVC, Atrial Fibrillation - AFib., etc).  We agree that a distinct advantage of synthetic data generation is the ability to create examples of rare events “at will”'; however, creating data that are faithful to real-world observations is non-trivial.   We did not simulate arrhythmia as we did not have a convincing PPG signal synthesis model that reflected these nuances. We do agree that one of the advantages of synthesized data is to create more “exotic” examples and oversample rare events. We hope that our research, and that of others, can build on this dataset in future to create more data with high quality renderings of rare physiological events. We were careful to not try to simulate events we did not feel confident we could create with a certain level of realism. We have added a “Future Directions” section to the paper to reflect this.
>
> The evaluation:
> We have added more results to the paper training on the SCAMPS dataset and testing on the widely used PURE dataset (Stricker et al., 2014). The results are consistent with those on the other datasets, while the model trained only on synthetic data performs well, it certainly leaves room for further exploration of the strengths and weaknesses of this synthetic dataset and its ability to generalize to all real-videos. Overall, given the results are consistent across all three datasets, we feel this provides a reasonably robust baseline measure of the performance.
>
> Thank you for the additional feedback regarding the QRS-complex and morphology modeling.  The references are very helpful and will hope to build on the level of nuance into our synthetics pipeline moving forward.

---

> > ### Comment · Reviewer_xdpX · 2022-08-26
> > **Comment on response**
> >
> > Thank you for adding Sec. 9 "Future Directions", and for the added evaluation in Tab. 2. These items further enhance an already important work.
> >
> > Re supporting arrhythmia from PPG, I appreciate the modelling problem and working out a route forward. I wonder if there is a way to draw on the experience of certain commercial vendors who have been accumulating data and releasing press articles to this end? Just a thought.
> >
> > ### Minor:
> > - line 319, missing period after ) ?

---

### Review · Ethics_Reviewer_tuBx · 2022-08-25

**Recommendation:** 1

**Ethics Review:**

The authors have addressed the potential ethical issues surfaced during the technical reviews and a further review of the paper raised no additional concerns.

---

### Meta-Review · Area_Chair_LBHD · 2022-09-02

**Recommendation:** Accept
**Confidence:** 5

**Metareview:**

The reviewers are positive regarding the high level of the contribution of the work for the NeurIPS 2022 Track Datasets and Benchmarks. The authors properly addressed all reviewers comments and concerns during the rebuttal period.

---

### Decision · Program_Chairs · 2022-09-16

Accept